# Phosphorylation of the Arginine-Rich C-Terminal Domains of the Hepatitis B Virus (HBV) Core Protein as a Fine Regulator of the Interaction between HBc and Nucleic Acid

**DOI:** 10.3390/v12070738

**Published:** 2020-07-08

**Authors:** Hugues de Rocquigny, Virgile Rat, Florentin Pastor, Jean Luc Darlix, Christophe Hourioux, Philippe Roingeard

**Affiliations:** 1Morphogenèse et Antigénicité du VIH et des Virus des Hépatites, Inserm–U1259 MAVIVH, Hôpital Bretonneau, 10 boulevard Tonnellé-BP 3223, CEDEX 1, 37032 Tours, France; virgile.rat@etu.univ-tours.fr (V.R.); florentin.pastor@etu.univ-tours.fr (F.P.); christophe.hourioux@univ-tours.fr (C.H.); philippe.roingeard@univ-tours.fr (P.R.); 2Laboratory of Bioimaging and Pathologies (LBP), UMR 7021, Faculty of Pharmacy, University of Strasbourg, 67400 Illkirch, France; jldarlix@gmail.com; 3Plate-Forme IBiSA des Microscopies, PPF ASB, Université de Tours and CHRU de Tours, 10 boulevard Tonnellé-BP 3223, CEDEX 1, 37032 Tours, France

**Keywords:** HBV, HBc, phosphorylation, packaging, RNA, assembly, chaperone activity

## Abstract

The morphogenesis of Hepatitis B Virus (HBV) viral particles is nucleated by the oligomerization of HBc protein molecules, resulting in the formation of an icosahedral capsid shell containing the replication-competent nucleoprotein complex made of the viral polymerase and the pre-genomic RNA (pgRNA). HBc is a phospho-protein containing two distinct domains acting together throughout the viral replication cycle. The N-terminal domain, (residues 1–140), shown to self-assemble, is linked by a short flexible domain to the basic C-terminal domain (residues 150–183) that interacts with nucleic acids (NAs). In addition, the C-terminal domain contains a series of phospho-acceptor residues that undergo partial phosphorylation and de-phosphorylation during virus replication. This highly dynamic process governs the homeostatic charge that is essential for capsid stability, pgRNA packaging and to expose the C-terminal domain at the surface of the particles for cell trafficking. In this review, we discuss the roles of the N-terminal and C-terminal domains of HBc protein during HBV morphogenesis, focusing on how the C-terminal domain phosphorylation dynamics regulate its interaction with nucleic acids throughout the assembly and maturation of HBV particles.

## 1. Introduction

Although a potent HBV (Hepatitis B Virus) vaccine is available, about 300 million people experience HBV infection, causing ~900,000 annual deaths due to liver-associated diseases, such as cirrhosis and hepatocellular carcinoma [1]. Antiviral treatments, including interferon and nucleoside/nucleotide analogues, are well tolerated, rarely leading to drug resistance. However, while these antiviral drugs suppress HBV replication, they do not cure viral infection due to the persistence of its genome in the nucleus of infected hepatocytes [2].

HBV is a hepadnavirus that packages its genome in the form of RNA, referred to as pre-genomic RNA (pgRNA) [3]. The ε stem-loop of pgRNA tightly interacts with the viral polymerase (Pol) formed by three domains, including the terminal protein (TP) linked by a spacer to the reverse transcriptase and the RNAse H domain [4]. Thus, Pol combines the properties of a reverse transcriptase (RT) and a DNA polymerase, allowing the conversion of pgRNA into relaxed circular (rc) DNA present in the interior of the infectious particle. The HBV infectious particles, named the Dane particles, consist of an envelope made of host lipids and three surface viral proteins (L, M and S) that are essential to promote the clathrin-mediated endocytosis of the virus through the sodium taurocholate co-transporting polypeptide (NTCP) pathway [5,6,7]. This envelope wraps a 30-nm diameter icosahedral capsid formed by the core protein (HBc) and ensures the protection of the 3.2 kb partially double-stranded relaxed circular (rc) DNA genome associated with the viral DNA polymerase [8]. Moreover, viral particle production upon the DNA transfection of hepatocellular carcinoma cells, from HepAD38 or HepG2.2.15 constitutively expressing the virus or from the serum of an infected patient reveals that most of the secreted particles are subviral particles devoid of any genome (Figure 1) [9,10] (for a recent review, see [11]). These particles correspond to self-assembled envelope proteins that are spheres or filaments [12,13]; these particles act as decoys for the immune system and are potent immunogens used for vaccine development [14]. The existence of other types of viral-like particles (VLPs) was shown to correspond to genome-free enveloped capsids or to RNA-containing capsids [15]. Note that genotype G was shown to produce HBsAg but failed to secrete subviral particles that are retained as aggregates in the cytoplasm [16].

The pgRNA is also a messenger RNA that is translated in HBeAg, HBcAg (HBc) and Pol. HBeAg (also referred to as precore protein) is a secreted protein that contains 29 residues upstream of the HBc sequence (Figure 2) thought to modulate the host immune response [17]. The HBc protein of the 183–185 residues is schematically divided into two domains (Figure 2), where the N-terminal domain (NTD; residues 1–140) is known to self-assemble in the capsid, linked by a flexible linker to the basic C-terminal domain (CTD; residues 150–183) that interacts with nucleic acids (NAs) [18]. This protein belongs to a large family of viral core proteins that are small and multifunctional. In addition to its structural role in the formation of capsid shells, HBc orchestrates the assembly of the viral particles through the recruitment of host and viral partners that are required for virus replication. HBc also interacts with NAs to ensure the protection of the viral genome and chaperone its replication, together with the viral polymerase. All these functions make this protein a target for the development of molecules endowed with anti-viral activities [19,20]. In this review, we will briefly summarize the role of the NTD in capsid formation and then focus on the role of the CTD at various steps of the viral cycle, notably how its phosphorylation regulates its interaction with nucleic acids. The importance of the CTD at other steps of virus replication, notably cell trafficking, genome release in the nucleus and HBV secretion, have recently been the subject of pertinent reviews [21,22].

## 2. The NTD Is Essential for Capsid Formation and Envelope Protein Recruitment

The structure of the NTD capsid has been solved by X-ray crystallography and cryo-electron microscopy, revealing the structures of monomers and dimers. Each monomer contains a series of five α helices (Figure 3). α1, 2 and 5 form a hydrophobic core that stabilizes the monomer structure, while amphipathic α3 and 4 participate in the dimer formation via a four-helix bundle, giving rise to the characteristic spikes on the surface of the capsid [24,27,28]. The interaction between dimers in the capsid involves residues of helix 5, in addition to the downstream proline-rich loop (residues 128–136) and the C-terminal arm (aa 137–142/3) [24,29,30]. This association between two dimers results in the formation of a large hydrophobic pocket where the Y132 residue is fully buried [24].

NTD alone can undergo self-assembly, forming a shell with two different icosahedral symmetries, namely T = 3 (minor assembly product 5%) or T = 4 (major assembly product 95%) (Figure 4) [31]. T = 4 means that the capsid is composed of 60 asymmetric units each made of two dimers of proteins (A–B and C–D), for a total of 240 capsid proteins per particle (Figure 4A–C). In the capsid formed of 240 monomers (T = 4), two dimers adopt an asymmetric orientation around the fivefold and twofold axes that triggers the formation of an icosahedral capsid (Figure 4D) [24]. Interestingly, the NTD structure exhibits large pores going from the outer to the inner layers that are about 12 to 25 Å in diameter depending on the axes from which they are observed in the icosahedral NTD capsid, by either X-ray [24] or cryo-EM [32,33,34] (for a review, [23]).

In addition to its role in the formation of the HBV capsid, the NTD is essential for envelope protein recruitment [30,35,36,37]. In fact, a series of residues mapping in the groove between the spikes and at the base of the spike was shown to be essential for the core to interact with L during the envelopment of the capsid [36,38]. Of note, the correlation between particle envelopment, found by Bruss et al. [36] using replication-competent particles, and the interaction between the core and L proteins in the absence of the HBV genome found by our group [38], favor the notion that the L-mediated egress of the HBV core is probably independent from core maturation. Thus, HBc–NTD is essential at several stages of the viral cycle and, consequently, is the target of an intense research for compounds endowing antiviral activity (for reviews, [19,20]).

## 3. The CTD Has Multiple Roles

A number of in vitro studies have shown that the NTD domain is necessary and sufficient for the formation of a capsid morphologically similar to capsids formed by the full length HBc. These studies led to the resolution of the NTD 3D structure and the identification of many parameters that govern capsid assembly [23]. Nevertheless, the NTD alone failed to assemble properly when purified from rabbit reticulocyte lysates, suggesting that the CTD is also required [39]. Moreover, our group recently showed, using fluorescence correlation spectroscopy (FCS), that HBc forms the capsid with an apparent Kd of 0.7–0.8 µM when expressed in Huh7 cells, while in this condition the HBc–NTD does not form capsids [7]. Thus, the CTD per se is not required for capsid formation but it facilitates the formation of the native shell capsid probably via RNA-binding as well as protein-protein interactions [33,39,40,41,42].

The CTD—which is highly conserved among all genotypes where substitutions are limited to the last five to seven residues (Figure 5)—exhibits two interesting features. Firstly, this short sequence of 34 (or 36) residues contains seven serines and one threonine, where seven of them could be phosphorylated (Figure 2, in red). As each phosphoryl group may carry one or two negative charges (see below), such a phosphorylation adds up to 14 negative charges per CTD (up to 14 × 240 = 3360 negative charges in the whole capsid). Secondly, the CTD contains 16 arginine residues (Figure 2, in green), forming a stable polycationic strand (16 × 240 = 3840 positive charges). Such a balance between the two sets of positive and negative charges is deemed essential for the proper regulation of the virus replication cycle.

### 3.1. Phosphorylation of the CTD (P-CTD)

#### 3.1.1. Most of HBc Phospho-Acceptors Are in the CTD

HBc is a phosphoprotein containing highly conserved phosphorylation sites located in the CTD (Figure 5). S155 (157), 162 (164) and 170 (172) are considered to be the major sites of phosphorylation [43,44,45,46,47,48]. Additional data identified a total of seven phosphorylation sites (S155, T160, S162, S168, S170 and S176) that are all in the CTD [42,49,50,51]. More recently, two serines (S44 and S49) amongst eleven S and twelve T residues in the NTD were also proposed as phospho-acceptors [52]. To further add to the complexity of amino acid phosphorylation, the phosphorylation of certain phospho-acceptors could enhance the phosphorylation of others [51,52], explaining, at least in part, why these sites play different roles in virus replication [42,43,47,50]. Taken together, these results propose that phosphorylation regulates electric homeostasis, which in turn drives pgRNA packaging, RT-directed synthesis of the rcDNA and capsid stability. Furthermore, the true number of negative charges is difficult to evaluate since three E residues are oriented toward the inner capsid (E46, E113 and E117) [53] and not all phospho-acceptors are phosphorylated at the same time (Figure 6) [50,52].

#### 3.1.2. Phosphorylation Can Be Mimicked by Acidic Residues (mP-HBc)

The role of each of these Ser/Thr residues was investigated by alanine substitution (UP-HBc) or by aspartic (D) or glutamic (E) replacement (mP-HBc). Alanine is neutral, while the D and E residues contain a carboxylate with a pKa of ~4. These residues are thus negatively charged at pH 7 and are considered to mimic the negative charge of the phosphate. Nevertheless, the phosphate group bound to S or T has either a single- or double-negative charge with a pKa of ~2 and ~7, which results in the mono- or bi-anionic state in the cellular environment [54]. Investigating as to whether the phosphate group is di-anionic or mono-anionic, notably during capsid assembly or during pgRNA packaging, would be essential for understanding the role played by phosphorylation/dephosphorylation in the course of virus replication.

#### 3.1.3. Kinase Identity

Several kinase candidates have been proposed to phosphorylate the core protein, such as kinase A, polo-like kinase 1 (PLK1), CDK2, kinase C alpha (PKCα) or SRPK [50,51,55,56,57,58]. This identification is challenging due to the high number of kinases and phosphatases in mammalian cells [59]. When *Escherichia coli (E. coli)* was co-transformed with two plasmids coding for HBc and SRPK1, seven phosphorylation sites, all located in the CTD, were characterized [42,49]. Such a high phosphorylation level, as compared to the medium one as previously reported (Figure 6) [46,47,48,50], was considered to result from the co-expression of both proteins and the use of a soluble SRPK1 truncated protein. It remains that the exact quantification of CTD phosphorylation during virus replication is difficult to evaluate in infected hepatocyte cells due to the dynamic nature of CTD phosphorylation, varying as a function of the phases of the cell cycle. This would explain, at least in part, the localization of the core protein [46,60]. In addition, the identity of the associated phosphatases has not been investigated yet.

### 3.2. Phosphorylation Is Not Essential for Capsid Assembly

If HBc phosphorylation occurs soon after translation, it takes place independently from capsid assembly, i.e., the HBc-Y132A mutant is deficient in capsid assembly but is phosphorylated [61,62]. Along the same line, the phosphorylation of the HBc-F97L mutant was found to be similar to wt HBc despite small modifications of the NTD structure [33]. When a HBc mutant with a substitution of all phospho-acceptors for E (mP-HBc) or A (UP-HBc) is expressed in Huh7 cells, no significant difference is observed for HBc expression and isolated capsids [50]. The assembly of HBc was at least not influenced by the mutation of newly identified phosphorylation sites of the NTD [52].

The low impact of phosphorylation on capsid assembly is in line with the formation of highly phosphorylated but empty viral particles obtained with a HBV genome defective for the expression of the polymerase or present in the serum of treated patients [15]. This is also in agreement with the RNA-independent protein–protein interaction-driven mechanism of capsid assembly obtained without the CTD [63] and to the poor effect of phosphorylation on capsid structure [33,42,56].

Nevertheless, wild-type capsids obtained from *E. coli* (UP-HBc) are less stable than capsids formed by mP-HBc where S155, S162 and S170 (see Figure 2, unbracketed numbers) were substituted for E [64]. This instability of wt capsids can be explained by the electrostatic repulsions of arginine residues, which are titrated by the glutamic acid residues in mP-HBc. As shown in Figure 6, HBc is always partially phosphorylated in a cellular context, as mimicked by mP-HBc in vitro. These observations favored the notion that the partial phosphorylation of CTD ensures a high stability of the HBV capsid.

## 4. The CTD Interacts with Nucleic Acids (NAs)

### 4.1. Role of Arginine Rich Domains (ARDs) in HBV Replication

The CTD sequence has been referred as to a protamine-like domain holding RNA or DNA binding facility [65,66]. It was thus proposed that the HBc–CTD has a histone-like function promoting DNA condensation [45]. CTD contains 16 R residues (Figure 2, green letters, I, II, III and IV) organized in four distinct domains ARDs [67], which are commonly found in RNA- and DNA-binding proteins (for a review, [68]). This arrangement is probably essential for different functions since it is conserved among human hepadnaviruses (Figure 5). Nevertheless, the first domain seems a little less preserved, with the replacement of R150 by other residues and of S155 by a threonine. In duck hepatitis B virus (DHBV), a model frequently used to characterize how hepadnaviruses replicate, the core protein is larger (262 residues) and contains the same basic residues, but they are scattered throughout the primary sequence [69].

Using partial CTD deletion or the substitution of R to A mutants, the four arginine stretches were originally proposed to have a specific role, notably, the first two arginine residues being more important for RNA binding and pgRNA packaging, while the two distal rows for DNA binding and reverse transcription [44,45,48,53,70]. For example, the truncated mutant HBc164, which contains the two first ARDs and one arginine of the third ARD cluster, efficiently packages shorter 2.2-kb spliced RNA rather than the entire wild-type 3.2-kb pgRNA [44,45,48]. Intriguingly, the 3′ end of the packaged pgRNA in the HBc164 capsid remained accessible to nuclease, suggesting that the 3′ end was protruding outside the capsid [48,71]. When HBc164 CTD was extended to arginine 173, a full length HBV genome was then obtained, while additional R residues (pos. 174, 175 and 179) were not required for DNA replication [48].

### 4.2. ARDs Endowed with NA Chaperone Activity

The relative importance of the ARDs at different stages of viral replication has been addressed by a thorough quantitative analysis of pgRNA packaging and (-) and (+)DNA synthesis, with mutants where the four clusters were substituted with A or K [72]. During the initial step of reverse transcription, where (-)cDNA anneals to the 3′ DR1 sequence [4], the alanine substitution in clusters II, III and IV impaired the minus strand template switch, causing a large decrease in (-)cDNA completion. Subsequent steps of DNA synthesis up to rcDNA completion were all affected at various levels by R–A substitution, I and II for (-)DNA elongation, III and IV for primer translocation, and all mutants had decreased DNA circularization and (+)cDNA elongation.

This allowed the group of Loeb to suggest that the core protein has nucleic acid chaperone (NAC) activity [72]. This NAC activity is well described as impacting the structure of abundant coding and non-coding RNA molecules, from prokaryotes to eukaryotes [73,74]. Briefly, this activity directs a transient destabilization of the NA structure and promotes the annealing of complementary NA sequences [75,76,77]. A few common properties allow for distinguishing RNA chaperones from other RNA-binding proteins: a lack of specificity for RNA or DNA, they are transiently needed, and there is no energy requirement for their activity. HBc does not share sequence similarities with other viral proteins, such as HIV-NCp7, HIV-TAT or HCV core protein, described as NAC factors, however, that all these proteins are highly basic and flexible so that they can adapt to the order–disorder transition according to their substrates [78]. A number of standard assays are used to monitor NAC activity, notably the binding, fraying and annealing of complementary sequences and the activation of the hammerhead ribozyme-directed cleavage of an RNA substrate [75,77].

The NAC activity of HBc was confirmed using DNA–DNA hybridization and hammerhead ribozyme cleavage in vitro [79]. Firstly, they monitored DNA annealing related to HIV-1 replication [80]—extensively used to follow NCp7, TAT and Gag NAC activity, Tar(+) and Tar(−) [76,81,82,83]—for duplex formation with increasing concentrations of assembled and disassembled capsids (Figure 7A). Interestingly, they monitored assembled (Figure 7A, lines 1–6) and disassembled HBc particles (Figure 7A, lines 7–11) and found that disassembled HBc was more efficient in facilitating duplex formation between Tar(+) and Tar(−). Note that in this assay, UP-HBc proteins were produced from *E. coli*. Using a series of HBc (147–183) peptides, they found that this chaperoning activity corresponded to the CTD sequence (Figure 7B). The partial deletion of ARD domains gave a significant decrease in Tar duplex formation but the relative importance of each ARD was not addressed. A kinetic analysis of HBc NAC would unravel by which pathway (kissing loop or stem) this duplex formation occurs [76].

### 4.3. Interaction of HBc with dsDNA

HBc was shown to interact in vitro with single- and double-stranded DNA, but the assembly of HBc onto dsDNA mostly led to the formation of large and heterogeneous complexes [84,85]. Nevertheless, HBc-dsDNA may be required for the formation of an HBc–cccDNA (covalently closed circular DNA) complex. Along this line, interaction may explain the localization of HBc in the nucleus. Bock et al. performed the immuno-precipitation of histones from the HBV-producing HepG2.2.15 cells and found histone proteins combined with both HBc protein and cccDNA [86,87]. These results were confirmed by transmission electron microscopy (TEM) with the identification of cccDNA decorated by HBc, showing a typical beads-on-a-string structure. Note that HBc was probably phosphorylated, because immuno-precipitation was performed from HepG2.2.15 extracts. Interestingly, they also found that HBc—monomeric but phosphorylated (from rabbit reticulocyte lysate)—interacted more tightly with HBV dsDNA as compared with HBV ssDNA or nonspecific DNAs [86]. Using in vitro-reconstituted HBV minichromosomes, they found that HBc molecules form an irregular chromatin organization with rcDNA while it was a regular chromatin assembly, notably with HBV dsDNA. The addition of HBc to the nuclear extracts of *Xenopus* oocytes (S150 extract), containing HBV DNA, induced the foreshortening of nucleosome spacing, suggesting that HBc directly participates in HBV cccDNA chromatin organization, which could regulate HBV transcription. This effect of HBc cccDNA organization was recently confirmed by chromatin immuno-precipitation (ChIP) assays [88]. The HBc-mediated pull-down of cccDNA was not affected when all R residues of cluster I were substituted for G. In contrast, the substitution of R for G in clusters III and IV drops the level of pulled-down cccDNA by a factor 10 and significantly lowers the level of HBV RNA, showing again that all R clusters are not equivalent in virus replication, notably during cccDNA transcription.

### 4.4. Specific Interaction between the CTD and NAs

As aforementioned, the negative environment during the first step of assembly in insect or mammalian cells prevents the nonspecific interaction with negatively charged cellular RNAs [15,42,48,53]. In the current model describing the recognition of pgRNA among the bulk of cellular RNAs, it is proposed that the newly synthetized capsid specifically recognizes the pgRNA–Pol complex [89]. This model would suggest that HBc exerts only a nonspecific interaction with NAs and does not actively participate in the direct packaging of its cognate RNA. Recent experiments revisited this relationship between capsid assembly and CTD-mediated pgRNA interactions [40]. First, a Systematic Evolution of Ligand by Exponential Enrichment (SELEX) approach identified three high-affinity binding sites for HBc dimers (Figure 8A, upper panel). These sites, referred to as preferred sites (PS1, PS2 and PS3), were proposed to act as packaging signals to specifically drive pgRNA packaging during HBc assembly. The importance of PSs for capsid formation was investigated by FCS. This technique monitors the fluorescence fluctuation of particles diffusing through the confocal volume, and the analysis of fluorescence fluctuation by an autocorrelation function allows the extraction of the hydrodynamic radius Rh (radius of the particle) [90]. Combining this highly sensitive technique and high-affinity RNAs allowed the authors to work at nanomolar range concentrations, which are supposed to be more relevant to in vivo conditions.

FCS data collected after the addition of 250 nM HBc dimers to a fixed concentration of labeled PSs were plotted as hydrodynamic radial distributions (Figure 8A, middle panel). This distribution, in addition to the shape of the capsid (Figure 8A, TEM images at the bottom) and to the sensitivity of particles to RNAse (not shown), was similar for the three PSs, suggesting that capsid morphogenesis was independent from this three PS sequences. In contrast, HBc assembly using PS-mutated RNAs or corresponding to epsilon or a DNA version of PS1 remained highly sensitive to RNAse, suggesting that the condensation of the nucleocapsid shell was specific to the RNA sequence and structure (not shown). Next, capsid assembly was followed without RNA (Figure 8A, grey line) or with unlabeled RNA (Figure 8A, blue line) and then fluorescently labeled. In the absence of RNA, HBc assembles predominantly in species with an Rh of 10–11 nm (gray), suggesting that the self-association of dimers is slow in the absence of PS RNAs, even though the low amount of assembled particles look like wild-type particles (Figure 8A, TEM images bottom right). The ability of HBc149 to interact with PS1 was assessed by FCS and the absence of an Rh increase clearly shows the absence of RNA-mediated HBc149 assembly, despite the formation of a HBc149 capsid observed by TEM (not shown). Thus, as expected, the truncation of the CTD does not hamper the formation of the capsid but is essential for RNA-dependent HBc assembly. Note that in this study, UP-HBc was purified from *E. coli*, and was thus highly positive. In this model using a low HBc concentration, UP-HBc alone forms few capsids (Figure 8B, upper panel). A higher concentration of HBc dimers is required for RNA-independent capsid assembly to bypass the repulsion of ARD positive charges. Thus, the absence of CTD (HBc149) or the reduction of charges by phosphorylation reduces repulsion forces, allowing the formation of particles at low concentrations (Figure 8B, middle panel). Conversely, in the presence of specific RNA, UP-HBc efficiently self-assembles (Figure 8B, lower row). Therefore, this study suggests that in vivo, at low concentrations, UP-HBc dimers probably recognize specific RNA structures similar to the structures identified here, which in turn would nucleate HBc assembly.

## 5. RNA Packaging Is Tuned by Phosphorylation and by the Nature of NAs

### 5.1. Effect of Phospho-Acceptor Modifications on RNA Packaging

UP-HBc capsids purified from *E. coli* are un-phosphorylated due to the absence of cognate kinases and being full of *E. coli* RNAs [42,45,49], while P-HBc capsids from insect and mammalian cells are empty of (host) RNAs if they are incompetent for pgRNA packaging [15]. UP-HBc capsids package RNAs 3400 nt in length close to the pgRNA size (3500 nt) [48,53]. The assembly of capsids expressed in Rabbit Reticulocyte Lysate (RRL), *E. coli* or 293T, where a few phospho-acceptors (three among seven) or all phospho-acceptors (seven) were changed to A (+ charges ↑), giving rise to PP-HBc and UP-HBc, resulted in the formation of capsids in a process dependent on the presence of host RNAs, concomitantly with a large decrease in pgRNA packaging [39,47,50,53]. These results are in line with the role of phosphorylation to prevent the packaging of cellular RNAs to the benefit of viral RNA.

In contrast, when these phospho-acceptors were changed to E/D (mP-HBc) (+ charges ↓) and the capsid was produced in RRL or 293T or when HBc was co-expressed with SRPK1 kinase in *E. coli* (+ charges ↓), giving rise to P-HBc, then capsid assembly was RNA independent, i.e., VLPs were found to contain a low amount of host RNAs [39,42,56]. Additionally, the double mutation of the S44–S49 in E (+ charges ↓) was also found to partially impair pgRNA packaging [52]. However, the two residues are not located in the CTD and probably do not directly interact with pgRNA. Interestingly, when ARDs were substituted for A (+ charges ↓), both pgRNA packaging and DNA synthesis were decreased [72], underlining the intimate balance between positive charges (arginine residues) and negatives charges (phosphorylation).

A scheme of this electrostatic equilibrium for RNA packaging is reported in Figure 9 [53]. The substitution of two or four arginine residues for AA in ARDs (+ charges ↓) proportionally decreases the size of the viral DNA and increases the amount of shorter packaged viral and spliced RNAs (Figure 9, VLPs 1 and 2). Of notes, in the quadruple mutant (AA substitution in the four ARDs), the height among the sixteen R residues remains in HBc, thus explaining the residual P-HBc-RNA interaction. Next, residues E40, E46, E113 and E117 located in the inner surface of the capsid shell and E180 located at the end of the CTD were substituted for A in double (or quadruple ARD) mutants (+ charges ↑) and a clear rescue of RNA was observed, suggesting that negatively charged glutamic residues participate in the titration of arginine positive charges (Figure 9, VLP 3). Similarly, when S155, 162 and 170 were substituted for A to mimic de-phosphorylation in the ARD mutants (+ charges ↑), it was found that a shorter DNA phenotype was rescued to produce longer DNA, as well as longer packaged RNA (Figure 9, VLP 4), in agreement with studies showing the strong impact of phosphorylation status on RNA packaging [43,47,72,91]. Altogether, electrostatic homeostasis plays an important role in the CTD–NA interaction that, in turn, regulates RNA packaging.

### 5.2. Phosphorylation Causes a Decrease in HBc-NA Interaction In Vitro

The ability of P-HBc extracted from Dane particles to bind HBV DNA in vitro was increased after de-phosphorylation (+ charges ↑) [84]. Conversely, when the HBc (147–183) peptide was phosphorylated at positions S155, 162 or 170 (+ charges ↓), a progressive inhibition of DNA annealing was observed and the phosphorylation state of these three serine residues completely abolished the DNA annealing activity (Figure 7C). The higher the serine phosphorylation of the HBc proteins was, the lower the DNA annealing activity and duplex formation. The treatment of the P-HBc (147–183) peptide by phosphatase restores the HBc (147–183) DNA-annealing activity (Figure 7C). In this assay, the effect of phosphorylation on the interaction between HBc and nucleic acids was monitored with non-specific substrates, favoring the notion that low-affinity complexes are formed. However, Stockley’s group also monitored the effect of phosphorylation on a more specific complex formed between HBc and PS1 [40]. To that end, HBc was phosphorylated prior to or during the interaction with PS1 (+ charges ↓) and in both cases P-HBc lost the PS1-binding activity but not capsid formation (Figure 8B). In the same vein, when S44 and S49, as potent photo-acceptors, were both substituted for E (+ charges ↓) in the NTD, a decrease in the interaction between HBc and rcDNA was observed [52].

### 5.3. Phosphorylation and RT Activity

HBV capsids, as well as DHBV capsids, purified from the cytoplasm are extensively phosphorylated, in contrast to virions purified from cell supernatant or from patient sera, suggesting that de-phosphorylation takes place during virion maturation [92,93]. In fact, intracellular and immature DHBV and HBV capsids were found heterogeneously phosphorylated, while mature capsids and virions were found dephosphorylated [94,95,96], even though cell-free particles contained P-HBc [42]. Nevertheless, the group of J. Hu reported that capsids of genome-free virions (referred to as empty virions) and naked capsids found in the cell culture supernatant or empty virions in the patient blood were extensively phosphorylated [96] (see a recent review in [11]). Taken together, these data suggest that the phosphorylation level is independent from virus production but rather relies on RT activity.

The correlation between RT activity and phosphorylation was first studied in the DHBV model. Using ultra-centrifugation methods to separate immature (RNA-containing) capsids, mature (dsDNA-containing) capsids and secreted virions, it was found that HBc in mature capsids and virions was dephosphorylated in contrast to immature capsids in which HBc was phosphorylated [95]. The link between de-phosphorylation and RT activity was also observed in the DHBV model, where the phospho-acceptors S (or T), substituted for D, impaired DNA synthesis [97] but not RNA packaging [98]. Likewise, when S was mutated for E in the HBV model, only a slight effect on pgRNA packaging was observed but single or double mutants containing E analogs hardly supported DNA replication [47,50]. More recently, the link between the HBc–RNA interaction and de-phosphorylation was demonstrated using the HBV model thanks to the use of antibodies specific for CTD phosphorylation [62]. Firstly, it was found that unassembled capsids were hyper-phosphorylated, in agreement with the phosphorylation status of empty capsids [96], while assembled capsids, produced from pgRNA encoding for the Y63F mutant DNA polymerase and containing only pgRNA, were mainly hypo-phosphorylated [62]. Taken together, these results suggest that, during the pre-assembly process, phosphorylation most probably causes a decrease in the affinity of capsids for cellular RNAs, and once assembly is kickstarted by the formation of the pgRNA–Pol complex, de-phosphorylation occurs in order to increase the HBc–pgRNA interaction and trigger RT activity through its NAC properties.

## 6. The CTD Localization during the Virus Replication Cycle

The structure of the CTD has not been solved yet but it is entirely possible that, due to its high content of basic residues, the CTD can adopt many different conformations. This conformational flexibility has indeed been found to be essential for modulating the CTD-mediated interaction of HBc with pgRNA in response to its phosphorylation status [34]. Interestingly, the position of the C-terminus in the native capsid has long been a matter of debate (Table 1). Early studies comparing native and disrupted core particles [99], or using proteinase digestion, have shown that the C-terminus faces the interior of the capsid [100,101,102]. This internal localization was also proposed by using radiolabeled HBc [103] and by measuring the binding to Ni^2+^ -NTA beads of NTD–His tags [104]. However, the requirements of the CTD during cytosolic transport, the anchorage of the capsid to the nuclear pore and nuclear import are not compatible with a permanent luminal localization of the arginine-rich domain [105,106,107]. Several observations favor the notion that this domain can protrude from the surface of the capsid.

CTD accessibility was investigated with different types of capsids produced in mammalian cells or *E. coli* [94]. For example, when maturated capsids (MatC) were purified, the unphosphorylated CTD was totally hydrolyzed by trypsin, in contrast to the phosphorylated CTD of immature capsids (Immat C) that were only partially hydrolyzed (see Figure 10). When UP-HBc capsids were obtained from *E. coli*, then the tight nonspecific interaction of *E. coli* RNAs prevented the UP-CTD exposure that becomes resistant to trypsin digestion. Interestingly, in similar conditions, when UP-HBc capsids were submitted to a denaturation–renaturation process (NA-free), the UP-CTD domain was partly exposed on the surface of the capsid [65,94] (Table 1). In conclusion, unspecific RNA traps the UP-CTD of *E. coli* (EcC) and specific RNA partially traps P-CTD inside the HBV capsid, while DNA excludes P-CTD, which is then dephosphorylated (MatC) (Table 1 and Figure 10). So, the RT-mediated conversion of pgRNA to dsDNA results in the exposure of the CTD on the surface of the capsid (see structure) [41].

In the work of Zlotnick and collaborators, capsids were produced in *E. coli* and RNAs were removed [64]. Wild-type UP-HBc and mP-HBc capsids were treated by trypsin and a clear accumulation of an HBc fragment corresponding to 1–157 was observed for both capsids [64], suggesting that, in both cases, the CTD protrudes from the surface of the capsid, as shown by cryo-EM [34]. Nevertheless, the CTD of wild-type UP-HBc was hydrolyzed faster than the CTD of mP-HBc, suggesting that the neutralization of CTD positive charges by EEE residues results in the stabilization of mP-HBc capsids. Similar results were obtained by comparing *E. coli*-derived UP-HBc_F97L and P-HBc_F97L sensitivity to trypsin treatment [42]. The structure of HBc_F97L is highly similar to that of wt HBc, with a small enlargement of the hydrophobic pocket in the spikes [33]. In UP-HBc_F97L capsids, the CTD was partly deleted while the CTD of P-HBc_F97L presented intermediate hydrolyzed products and was then entirely hydrolyzed. Note that in this work, UP-HBc contained large amounts of bacterial RNAs, while P-HBc was RNA-free, in line with the decrease in HBc–NA affinity when HBc is phosphorylated. In line with these results, UP-HBc (without RNA) capsids were able to interact with SRPK1, a SR protein kinase specific for CTD, while neither HBc149 nor RNA-filled UP-HBc showed SRPK1 binding, emphasizing that in empty and UP-HBc capsids, the CTD is protruding outside the capsid [56] (Table 1).

This flip from the interior to the exterior of the capsid is consistent with the flexibility of the linker peptide (residues 143–149) and to numerous pores on the capsid surface [24,42,110]. By cryo-EM, the pores of empty and UP-HBc capsids appear partially occluded by the surface-exposed CTD, while in pgRNA-containing UP-HBc particles, pores remain accessible [34]. In contrast, the structure of full-length HBc capsids purified from *E. coli* and full of *E. coli* RNAs, obtained at a 3.5Å resolution by cryo-EM, revealed the exposure of the CTD at the capsid surface [108]. The nature of the nucleic acid involved in the regulation of CTD exposure is important. As a matter of fact, the affinity of CTD for bacterial RNA may be lower than for pgRNA, which may explain why the CTD of a capsid full of unspecific RNAs may float on the surface of the capsid, while the CTD of a capsid harboring specific pgRNA may be sequestered within the capsid.

## 7. Summary of the Role of the CTD Phosphorylation during HBV Morphogenesis

The initial step of HBV morphogenesis only necessitates the HBc–NTD that self-assembles via protein–protein interactions (Figure 10, ①, ②) [24,111]. Concomitant with HBc–NTD assembly, the recruitment of a kinase takes place [40,51,53,64] (Figure 10, ②). Up to seven phospho-acceptor sites located in the CTD and possibly two in the NTD could be phosphorylated (red bars). However, only some of them appear sufficient for pgRNA packaging [48,50,51,52,53]. CTD phosphorylation occurs within the shell by a capsid-trapped kinase [112] or together with oligomer formation, with, however, no influence per se on capsid shape [33,34,42,61,62], but rather on the kinetics of assembly [7,40]. This phosphorylation regulates the electrostatic homeostasis of this molecular assembly, thus preventing the electrostatic repulsion of the arginine positive charges. HBc-CTD then might be oriented predominantly inside the nascent capsid. This process appears to be self-controlled since the HBc–SRPK1 complex responds to the degree of phosphorylation [56]. At this step of the replication cycle, the HBc–CTD might not be fully phosphorylated (green bars in immature capsids) but it may be enough to prevent the packaging of cellular RNAs and trap the CTD domain in the interior of the nascent capsid [53,64,94] (Table 1).

It is likely that, at this stage of the viral cycle, two assembly pathways could be distinguished. The first assembly path occurs at a low concentration of HBc, where a subpopulation of unphosphorylated dimers (UP-HBc in green) specifically interact with pgRNA (Figure 10, ③) [40]. When pgRNA translation takes place on the polysome machinery, Pol and HBc are synthesized at the same time, greatly facilitating local RNA–protein interactions mediated by their respective high-affinity binding sites [40,89,113,114]. This results in the specific packaging of the viral RNA in a replication-competent capsid [115]. Consistent with this view, pgRNA serves as a platform directing the HBc–HBc interactions and condensation necessary for capsid completion [116]. Even though RNA-containing particles could be secreted [11] (Figure 10, ④) it is likely that, concomitant with capsid formation, RT, assisted by the nucleic acid chaperone activity of HBc, starts the synthesis of the (-)cDNA strand (Figure 10, ⑤) [72,79]. The formation of dsDNA (rcDNA), also referred to as the maturation process, increases the number of negative charges. This induces a repulsion of the P-CTD (red bars) [34,64,85] that is then exposed to the surface of the capsid thanks to the flexibility of the linker [42,110] and to the multiple holes in the capsid shell [24,32,33,34,84,94] (Table 1). Alternatively, taking into account that CTDs are heterogeneously phosphorylated, it is possible that a population of P-CTDs remains outside of the capsid during assembly and, conversely, only the UP-CTD ones, interacting with the pgRNA, remain within the capsid. Regardless of the model, mature viral particles are characterized by the exposure of several P-CTDs extruded out of the capsid that might be partly dephosphorylated (green bars), in agreement with the unphosphorylated status of complete secreted virions (Figure 10, ⑥) [56,95,96]. For non-secreted particles, their phosphorylated nuclear localization signal (NLS) located in the CTD is then used to dock mature (or immature) capsids on nuclear pores (NPCs) (Figure 10, ⑦) of the nuclear envelope, notably through the importins α and β [107], where only the mature capsid is disassembled [22] (Figure 10, ⑧). The brittleness of the maturated capsid induced by the rigidity of the DNA [41,85] and its phosphorylation status would direct particle dissociation so as to release the viral genome into the host cell nucleus [52,94,117], where cccDNA formation occurs, establishing chronic infection [8]. The second assembly pathway could occur at higher protein concentrations as an RNA-independent assembly pathway (Figure 10, ⑨). The gradual increase in protein concentration compensates for the weak protein–protein interactions [63], allowing the formation of large amounts of empty P-HBc particles [96]. Such empty capsids with exposed and phosphorylated CTDs (Table 1) can then be transported to the nucleus [22,105] but through an importin-β-dependent pathway (Figure 10, ⑩) [118] or secreted as naked or enveloped particles (Figure 10, ⑪) [11].

In conclusion, the CTD has pleiotropic functions that take place during the initial step of HBV morphogenesis, which are all tightly regulated by the electrostatic homeostasis carried out by serine and threonine phosphorylation that counterbalances ARD motifs.

## Figures and Tables

**Figure 1 viruses-12-00738-f001:**
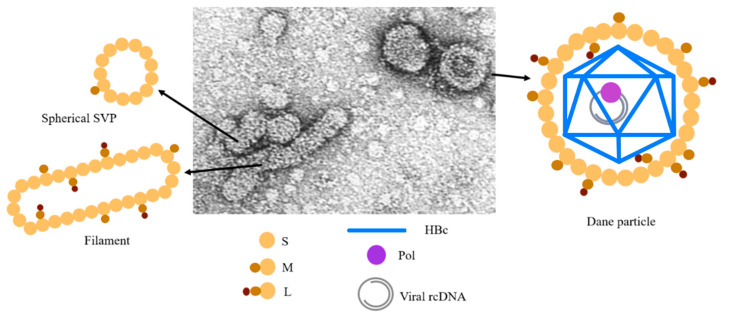
Schematic representation of HBV virions. Spherical and filamentous particles (SVPs) correspond to small (20 nm) entities only composed of the envelope proteins [13]. They are empty, not infectious and represent the most abundant particles found in the supernatant milieu of infected cells or in the sera of chronically infected subjects. Infectious particles (Dane particles) are larger, 42 nm in diameter, and are formed by an icosahedral capsid embedded in a lipid membrane containing the S, L and M surface proteins [10]. The internal nucleocapsid contains the circular partially double-stranded genomic DNA (rcDNA), about 3.2 kb in length, which is covalently linked to the viral reverse transcriptase. More recently, a large quantity of empty virions corresponding to enveloped capsid without rcDNA were found in cell supernatant or patient serum [11].

**Figure 2 viruses-12-00738-f002:**
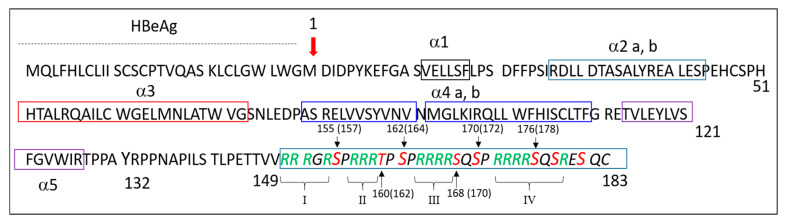
HBc primary sequence. HBc sequence starts at position M1. The upstream 29 residue sequence corresponds to a peptide signal that drives the translocation of the 25 kDa precore protein into the ER lumen, leading to the synthesis of a secreted form of the core protein, named HBeAg [23]. The N-terminal domain (NTD) corresponds to residues 1–140. Rectangles delineate the five alpha-helices of NTD found in NTD monomers that are essential for HBc to assemble in the icosahedral capsid [24]. The next ten residues contain the PPXA sequence found in various viral proteins to hijack the Endosomal sorting complexes required for transport machinery (ESCRT) [25,26]. This flexible sequence was recently confirmed to be essential during HBV replication. The C-terminal domain (CTD) contains arginine residues (in green) required for the interaction of HBc with nucleic acids (NAs), an activity that is regulated by the phosphorylation of the seven serine/threonine residues (in red), even though S 155, 162 and 170 are considered as the three major phosphorylation sites (residue numbers of S or T in brackets, genotype A).

**Figure 3 viruses-12-00738-f003:**
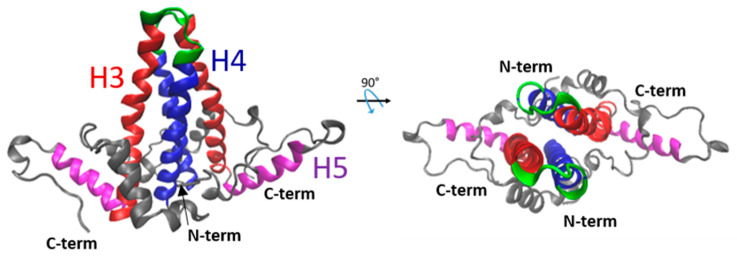
Representation of the 3D structure of the HBc–NTD dimer. Two orthogonal views of the HBc dimer are shown. This dimer structure was obtained from the Protein Data Bank (PDB) 1QGT [24]. Each monomer is formed of five α-helices. Helices 3 (red) and 4 (blue) display a hairpin shape assembling into a four-helical bundle within the dimer, generating the capsid spikes of the particle. The spike tip (in green) corresponds to the highly immunogenic c/e1 epitope [23]. The fifth helix (magenta) and the downstream proline-rich loop are involved in the dimer–dimer oligomerization.

**Figure 4 viruses-12-00738-f004:**
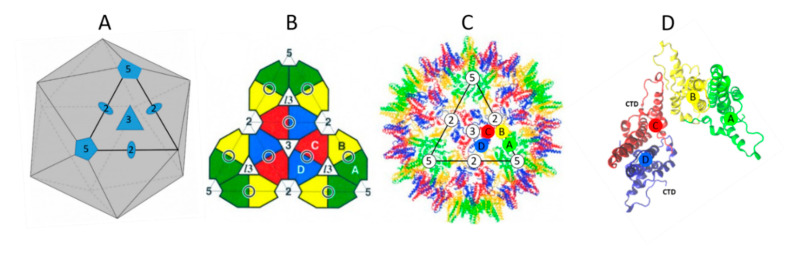
Dimer assembly in the T = 4 capsid. The HBV capsid structure was generated from cryo-electron microscopy (cryo-EM) analyses and X-ray crystallography (XR) with HBc proteins produced in vitro or from HBV-infected persons [23]. (**A**): Scheme of an icosahedron (adapted from [32]). It contains 20 facets (triangle) with the threefold axes going through the center of the opposite face, twelve fivefold axes (pentagon) going through the opposed vertices and fifteen twofold axes (oval) passing though the centers of the opposite vertices. (**B**): View of one icosahedron face through the threefold axis [24]. A, B, C and D correspond to 2 × 2 dimers and the central black ring corresponds to the four helix bundles. Subunit nomenclature and color were used according to [27,32]. (**C**): Structure of HBV capsid (T = 4) viewed from the threefold axis [24]. (**D**): A–B and C–D HBc dimers oriented to form the T = 4 capsid. These structures were obtained from the PDB, file 1QGT, and colored by using visual molecular dynamics (VMD) 1.9.3.

**Figure 5 viruses-12-00738-f005:**
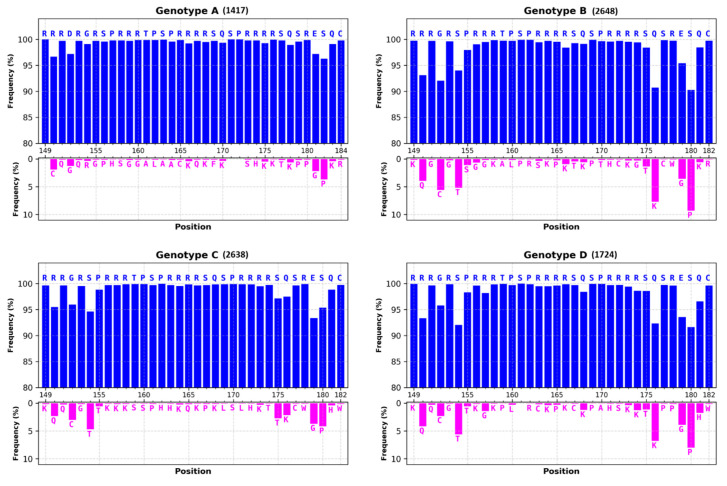
Conservation of the CTD amino acid sequences of the most prevalent genotypes. CTD sequences were extracted from the HBV database at Institut de Biologie Physico-Chimie (IBCP) IBCP (Lyon, France, https://hbvdb.ibcp.fr/HBVdb/). Numbers in parentheses correspond to numbers of analyzed sequences. Blue histograms correspond to overall conservation of residue at each position of the consensus sequence presented at the top of the graph. Pink histograms correspond to the most frequent substitution found in sequence alignment. Note that the genotype A CTD possesses two extra amino acids. Vertical axes represent frequencies of amino acid residues, expressed as percentage of conservation. In blue histograms, the axis was normalized between 80 to 100%, while in pink histograms, the axis was normalized between 0 to 10%, except for genotype E.

**Figure 6 viruses-12-00738-f006:**
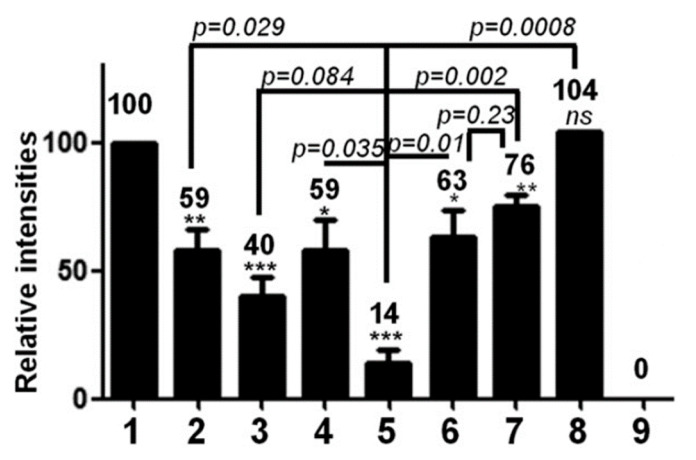
Not all S or T phospho-acceptors are phosphorylated at the same time. This histogram represents the relative intensities of phosphorylated HBc and HBc derivatives. Huh7 cells were transfected with a plasmid expressing wt HBc or HBc derivatives, where S or T were replaced with A and incubated with [^32^P] orthophosphate. (1) wt HBc STSSSS corresponds to the wild type (see Figure 2, numbers in brackets), (2) All serines were substituted for A while T162 was not (mut. ATAAAA), (3) S170 was maintained (mut. AAASAA), (4) S178 was maintained (mut. AAAAAS), (5) All phospho-acceptors were substituted for A (mut. AAAAAA), (6) S157, 164, 172 were substituted for A (mut. 3A-PRRR), (7) T162, S170 and S178 were substituted for A (mut. RRR-3A), (8) T162 was substituted for S (mut. SSSSSS), (9) pcDNA as control. Two days post transfection, HBc was immune-precipitated with anti-HBc antibody, and loaded onto SDS-PAGE and autoradiographied. Note that the presence of a single S (wells 3 and 4) or T (well 2) is enough for an efficient phosphorylation of HBc proteins. These results suggest that in the wt HBc only a few potent phospho-acceptor sites are phosphorylated. Modified from Figure 2B of [50] (IDccc: 1035507-2). Asterisk symbolizes a statistical difference between two samples.

**Figure 7 viruses-12-00738-f007:**
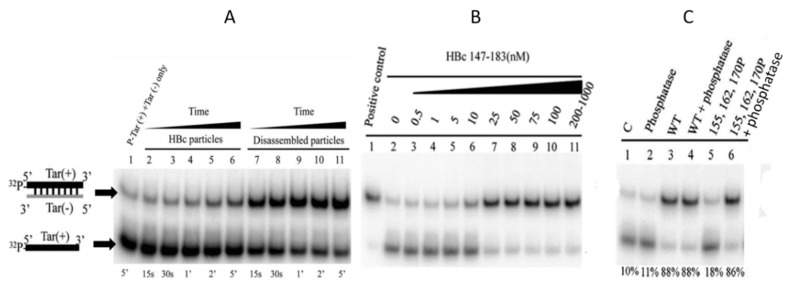
HBc and HBc (147–183) nucleic acid chaperone activity. (**A**): Tar(+)/Tar(−) duplex formation by assembled and disassembled capsids. (**B**): Tar(+)/Tar(−) duplex formation mediated by HBc (147–183) peptides. (**C**): Phosphorylation modulates Tar(+)/Tar(−) duplex formation. Tar(+) and Tar(−) were incubated with: Line 3, UP-HBc (147–183); Line 4, HBc (147–183) treated with protein phosphatase; Line 5, P-HBc (147–183) at positions S 155, 162 and 170; Line 5, P-HBc (147–183) at positions S 155, 162 and 170 treated with protein phosphatase. Panels A, B and C were respectively derived from Figures 1, 3 and 5 of [79] (IDccc: 1035507-1).

**Figure 8 viruses-12-00738-f008:**
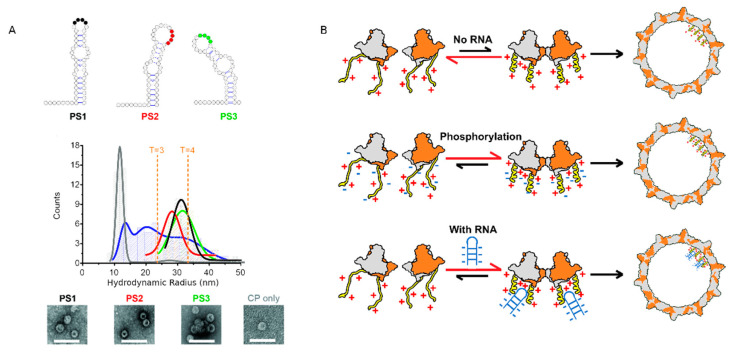
High-affinity binding site (preferred site, PSs) trigger sequence-specific Viral Like Particle (VLP) assembly. (**A**) Dye end-labeled RNA oligos, encompassing PS1 (black), PS2 (red) or PS3 (green), were each assessed for their ability to form VLPs at nanomolar concentrations using single molecule Fluorescence Correlation Spectroscopy (smFCS). PS oligonucleotides were present at a concentration of 15 nM. FCS data were collected in 30 sec bins, and data collected after the addition of 250 nM HBc dimers (taken to be the endpoint of assembly) were then plotted as hydrodynamic radial distributions (middle panel). Unlabeled RNA oligos, encompassing PS1 (blue) and HBc alone (gray), were also assessed for the ability to form VLPs at identical concentrations. At the end of these reactions (after the addition of 250 nM Hbc dimers), HBc was labeled with Alexa Fluor-488 and FCS data were collected in 30 sec bins for 100 min. The resulting R_h_ distributions were plotted as above. Note, the dye-labeling of the HBc dimers prevents them from assembling, implying that this is an end-point measurement. The successful assembly of T = 4 VLPs results in a labeled species of R_h_ ~33–38 nm, as seen by TEM, in assemblies with PS oligonucleotides (lower panel). HBc only assembles predominantly in species with an R_h_ of 10–11 nm (gray), suggesting that the self-association of dimers is slow in the absence of PS RNA. (**B**) Model for CTD-mediated HBc assembly. NTD dimers are in gray and orange, CTD in yellow and the RNA in blue. A kind gift from N. Patel and P.G. Stockley.

**Figure 9 viruses-12-00738-f009:**
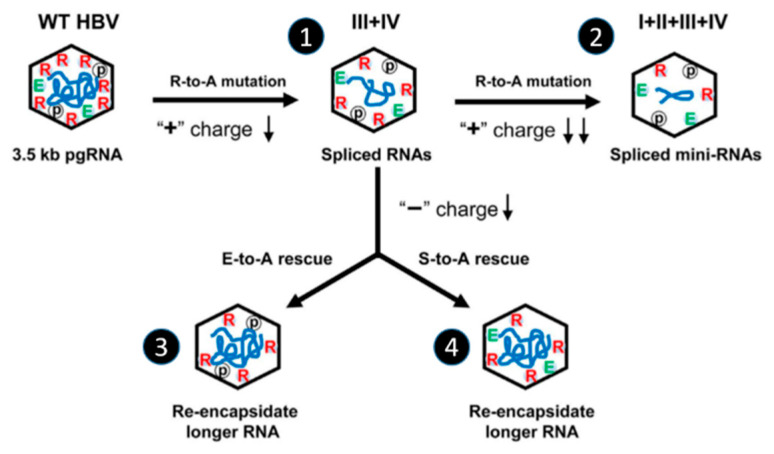
A subtle charge balance is required for RNA packaging. ARDs were mutated (R substituted for A) to reduce the positive charges and the status of packaged RNA is schematically shown by the blue line (pgRNA, 2.2-kb spliced RNA, 1.5 spliced mini-RNA). III + IV corresponds to HBc proteins where R165, 166, 173 and 174 of the third and the fourth ARDs were substituted for A. I + II + III + IV corresponds to HBc protein where R151, 153, 157, 158, 165, 166, 173 and 174 of all ARDs were substituted for A. The lower amount of positive charge of III + IV can be rescued by E–A or S–A substitution. This result suggests that the relative ratio of positive to negative charge appears more important than the quantity of either positive or negative charges for RNA packaging. Extracted and modified from Figure 7 of [53] and published under a CC BY 4.0 license.

**Figure 10 viruses-12-00738-f010:**
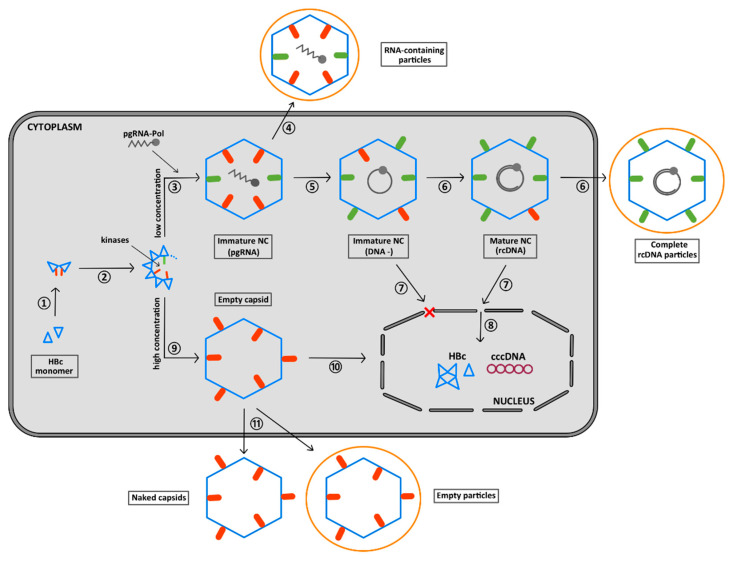
Phosphorylation and CTD exposure during HBV replication. CTD localization is presented inside and outside capsids as small rounded bars. Phosphorylated CTDs are in red, while unphosphorylated CTDs are in green. Numbers correspond to the different steps during which phosphorylation and/or CTD flips possibly occur.

**Table 1 viruses-12-00738-t001:** Different localizations of the CTD.

		Phosphorylated	Unphosphorylated
RNA	pgRNA	In (partial) [94] In [34]	In [34]
	Unspecific RNA	In (partial) [94] In [56]	In [94,103] In (partial) [42,108]
DNA		Out [84]	Out [94,95,109]
Empty		Out (partial) [42,56,65]	Out (partial) [34,56,64,65]

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
