# Peer review of "Phosphorylation of the Arginine-Rich C-Terminal Domains of the Hepatitis B Virus (HBV) Core Protein as a Fine Regulator of the Interaction between HBc and Nucleic Acid"

_viruses, 2020, doi:10.3390/v12070738_

Round 1
Reviewer 1 Report
This is a very well written and informative review that summarizes a lot of current information about the HBV capsid with a focus on its C-terminal domain.
There are just a few minor points that could be adressed:
-line 47, 47 HBV genotype G fails to release significant amount of subviral particles
line 166: references should be included for some papers describing in vitro assembly of HBV capsids
line 168: more details for the fluorescence based determination of the Kd for capsid formation should be provided in the text
line 234: in addition to kinases involved in phosphorylation of core the authors should provide a short overview about phosphatases involved in dephosphorylation of core.
line 360: a more careful description about the phosphorylation by reticulocyte lysate should be given. The kinase pattern in reticulocyte might be different from the kinase pattern in hepatocytes.
line 509: informations about the phosphorylation of released naked capsids should be included.
Author Response
Reviewer 1
This is a very well written and informative review that summarizes a lot of current information about the HBV capsid with a focus on its C-terminal domain.
Thank you for your comment.
There are just a few minor points that could be addressed:
-line 47, 47 HBV genotype G fails to release significant amount of subviral particles
In fact HBsAg secretion is impaired for genotype G. A sentence and a ref ‘’Peiffer et al, Journal of Hepatology, 2015’’ (N°16) were added (line 58).
line 166: references should be included for some papers describing in vitro assembly of HBV capsids
In fact ref 23‘’ Venkatakrishnan B & Zlotnick A. (2016). The Structural Biology of Hepatitis B Virus: Form and Function. Annu Rev Virol, Vol: 3, p: 429-51, ID: 10.1146/annurev-virology-110615-042238.’’’ was added (line 151).
line 168: more details for the fluorescence based determination of the Kd for capsid formation should be provided in the text
This Kd was determined using Fluorescence Correlation Spectroscopy (FCS) (REF 7). This technique measures the intensity fluctuations of fluorescent species within a femtoliter volume. Analysis of such fluctuations using Photon Counting Histogram method distinguishes core particle oligomers from monomers. Taking into account the volume of the measurements, an assembly binding curve was then constructed to determine the apparent Kd for HBV capsid assembly. These details are highly technical. It seems not necessary to add them in the text (line 153).
line 234: in addition to kinases involved in phosphorylation of core the authors should provide a short overview about phosphatases involved in dephosphorylation of core.
Unfortunately, up to my knowledge, no data on the role of phosphatase in the core dephosophorylation have been found in the literature.
line 360: a more careful description about the phosphorylation by reticulocyte lysate should be given. The kinase pattern in reticulocyte might be different from the kinase pattern in hepatocytes.
In fact, the kinase pattern of highly differentiated cells like reticulocyte might be different from the pattern kinase of de-differentiated HepG2.2.15 cells. Nevertheless, in Bock et al (now ref N°87), the aim was not the identification of kinase but the effect of HBc on HBV minichromosme architecture.
line 509: informations about the phosphorylation of released naked capsids should be included.
Phosphorylation status of the naked capsid was in fact included (ref 97). This sentence (line 455 in the revised text) was modified since the phosphorylation of naked capsid was described only with capsids isolated from the cell culture, not from human serum. In addition, ref 14 (Ning et al, Plos Patho, Vol 7, 2011) was removed. In this work they point out the secretion of genome free virions but no correlation between empty virion secretion and the CTD phosphorylation status was established. At least, phosphorylation status of empty and naked capsid is now schematically presented in the bottom of Fig. 9 (third reviewer).
Reviewer 2 Report
Here Rocquigny et al. review the functions of phosphorylation in the assembly, maturation and transport to the nucleus of the hepatitis B virus nucleocapsid. The field is rich, dense and complex. Unfortunately the review suffers from several shortcomings that detract from its helpfulness and should be corrected before publication.
1) My main problem is that the review should be much more concise. Several papers are described in fine detail instead of being summarized. This includes reproducing figures in whole or in part and describing them as in the results section of an original paper (but without the methods section). All such figures should be removed: Figures 5, 6, 7, 8, 9, 10, 11 (except panel D, which is a real summary). Instead the authors should focus on summarizing cartoons in the figures (for instance adapting Figure 9B) and giving the gist of the papers, including why some results seem to contradict others (for instance CTD accessibility depending on phosphorylation but also capsid (lack of) content or addition of a destabilizing partner such as importin beta).
2) The summarizing final section "VII - Summary of the role of the CTD phosphorylation during HBV morphogenesis" actually contradicts the previous sections: l. 647 "Regardless the model, processed viral particles are characterized by the exposure of several P-CTDs extruded out of the capsid [55, 96, 97]." after recounting that reverse transcription is linked to dephosphorylation.
3) line 501 This implies that while phosphorylation is critical in driving the specific packaging of pgRNA diluted in the bulk of cellular RNAs, it is essential that de-phosphorylation can take place to increase HBc-NA interactions in order to promote RT activity and convert the pgRNA to rcDNA.
The sentence is misplaced. What comes before ("This") has nothing to do with RT activity and its link to dephosphorylation, that are examined only in the next section.
4) line 372 IV - 3 - Specific interaction between the CTD and NA.
Here the authors give a lengthy account of (Patel et al., Nat Microbiol 2017) (their ref 39) but forget to actually cite it. A shorter account should be given without figure 9 (point 1 above).
5) The authors do not examine the possible functions of phosphorylation in the nucleus. The title should be changed.
6) I suggest several corrections to avoid confusion:
line 78: sequence corresponds to a peptide signal that drives the ER membrane insertion of the 25kDa precore protein -> sequence corresponds to a peptide signal that drives translocation of the 25kDa precore protein into the ER lumen
line 90: The structure of the NTD monomer and dimer has been solved by X-ray crystallography and cryo-electron microscopy. Data obtained reveal that each monomer -> The structure of the NTD capsid has been solved by X-ray crystallography and cryo-electron microscopy, revealing the structures of monomer and dimer. Each monomer
line 92 form a hydrophobic pocket that stabilizes the monomer structure -> form a hydrophobic core that stabilizes the monomer structure
line 94 Homodimer interaction involves -> Interactions between dimers in the capsid involve
line 185 Such a phosphorylation adds up to 14 negative charges per CTD (14×240 =3360 negative charges) -> As each phosphoryl group may carry one or two negative charges (see below), such a phosphorylation adds up to 14 negative charges per CTD (up to 14×240 =3360 negative charges in the whole capsid)
line 350 the assembly of HBc onto dsDNA did not account for formation of large and heterogeneous complexes [85, 86].
Do the authors mean "the assembly of HBc onto dsDNA led mostly to large and heterogeneous complexes" ?
7) line 116 Figure 4: Panel B is taken from ref 21. This should be referenced in the legend. Are C and D original panels ?
8) Small errors
ref 37 and 38 are the same
line 256 the poor effect of phosphorylation on capsid structure [30, 41, 55, 63]. Ref 63 is not relevant to phosphorylation.
9) Typos
41 S) essential to promote clatherin mediated endocytosis
142 hepatraocellular
155 (upper raw)
160 corresponding to c+v+ (green), c+v- (bleu)
210 S178 was maintained, 5- AAAAAA all photo-acceptors
306 IV - 1 - b ARDs endowed with NA chaperon activity
314 mutants with a decreased
332 8A, lines 7-11) and found that dissembled HBc
477 height among the sixteen R residues remains in HBc
Author Response
Reviewer 2
1) My main problem is that the review should be much more concise. Several papers are described in fine detail instead of being summarized. This includes reproducing figures in whole or in part and describing them as in the results section of an original paper (but without the methods section). All such figures should be removed: Figures 5, 6, 7, 8, 9, 10, 11 (except panel D, which is a real summary). Instead the authors should focus on summarizing cartoons in the figures (for instance adapting Figure 9B) and giving the gist of the papers, including why some results seem to contradict others (for instance CTD accessibility depending on phosphorylation but also capsid (lack of) content or addition of a destabilizing partner such as importin beta).
‘’’ All such figures should be removed ‘’’ Some figures were removed (5, 7, 11) or partially deleted (6 now n°5 and 10, now n°8). To illustrate our text, figures 8 (now n° 6) and 9 (now n°7) were conserved. The panel D of fig 11 was redundant with Table 2 and as a consequence was removed as well. Legends of fig 5 (lines 191-203) and fig 8 (410-418) were shortened. As fig 5, 7, and 11 were removed chapters II (line 137), IV (line 259) and VI (line 491) were shortened. A new fig (fig. 9) was added as recommended by the third reviewer.
‘’’’CTD accessibility depending on phosphorylation but also capsid (lack of) content’’’ The CTD accessibility as function of phosphorylation and/or nucleic acid is discussed (§ VI, line 478), summarized in table 2 (line 503) and schematically presented in fig. 9 (line 545).
“””destabilizing partner such as importin beta”” The effect of importin on capsid destabilization was not evocated in this review. People interested in the role of the CTD on HBc trafficking are invited to read two reviews n° 21 and 22. This review is more focused on the role of the phosphorylation on HBc-NA interaction. This point was clarified (lines 82-84) and the title of the review modified (Line 4). Nevertheless, a few precisions were added in the summary ending this review to take into account this comment (lines 569-583).
2) The summarizing final section "VII - Summary of the role of the CTD phosphorylation during HBV morphogenesis" actually contradicts the previous sections: l. 647 "Regardless the model, processed viral particles are characterized by the exposure of several P-CTDs extruded out of the capsid [55, 96, 97]." after recounting that reverse transcription is linked to dephosphorylation.
This sentence was corrected. The exposition of the P-CTD during the particle maturation induces the de-phosphorylation of the CTD (Line 572).
3) line 501 This implies that while phosphorylation is critical in driving the specific packaging of pgRNA diluted in the bulk of cellular RNAs, it is essential that de-phosphorylation can take place to increase HBc-NA interactions in order to promote RT activity and convert the pgRNA to rcDNA.
The sentence is misplaced. What comes before ("This") has nothing to do with RT activity and its link to dephosphorylation, that are examined only in the next section.
In fact, in this § the relationship between RT activity and dephosphorylation was not presented. This § presents a correlation between the phosphorylation status and HBc-NA interaction. Phosphorylated proteins (P-CTD) interact weakly with NA while non-phosphorylated proteins (UP-CTD) interact strongly with NA. This sentence was removed. The correlation between RT activity and the affinity of HBc for NA mediated by the phosphorylation status is summarized at the end of § V-3 (lines 474-477).
4) line 372 IV - 3 - Specific interaction between the CTD and NA.
Here the authors give a lengthy account of (Patel et al., Nat Microbiol 2017) (their ref 39) but forget to actually cite it. A shorter account should be given without figure 9 (point 1 above).
In fact ref 41 was added in the text (line 339). Figure 9 (now Fig 7) was left and the text was not shortened. This work is very important to understand how the CTD phosphorylation is essential to discriminate unspecific and specific RNA.
5) The authors do not examine the possible functions of phosphorylation in the nucleus. The title should be changed.
In fact, this review is focused on the role of phosphorylation on the interaction between HBc and NA in the cytoplasm. The title was changed as suggested by this comment –first issue- (line 4).
6) I suggest several corrections to avoid confusion:
line 78: sequence corresponds to a peptide signal that drives the ER membrane insertion of the 25kDa precore protein -> sequence corresponds to a peptide signal that drives translocation of the 25kDa precore protein into the ER lumen
This sentence was corrected (line 90)
line 90: The structure of the NTD monomer and dimer has been solved by X-ray crystallography and cryo-electron microscopy. Data obtained reveal that each monomer -> The structure of the NTD capsid has been solved by X-ray crystallography and cryo-electron microscopy, revealing the structures of monomer and dimer. Each monomer
This sentence was corrected (line 101)
line 92 form a hydrophobic pocket that stabilizes the monomer structure -> form a hydrophobic core that stabilizes the monomer structure
‘’pocket’’ was replaced by ‘’core’’ (Line 103).
line 94 Homodimer interaction involves -> Interactions between dimers in the capsid involve
This sentence was corrected (line 105)
line 185 Such a phosphorylation adds up to 14 negative charges per CTD (14×240 =3360 negative charges) -> As each phosphoryl group may carry one or two negative charges (see below), such a phosphorylation adds up to 14 negative charges per CTD (up to 14×240 =3360 negative charges in the whole capsid)
This sentence was corrected (line 171)
line 350 the assembly of HBc onto dsDNA did not account for formation of large and heterogeneous complexes [85, 86].
Do the authors mean "the assembly of HBc onto dsDNA led mostly to large and heterogeneous complexes" ?
Yes!. This sentence was corrected (now line 312).
7) line 116 Figure 4: Panel B is taken from ref 21. This should be referenced in the legend. Are C and D original panels ?
Panels B and C were taken in ref 24. This ref was added for panel B (line 132). In C a triangle between five-fold and two-fold axes was drawn. In addition, dimers AB and CD were reported with the same color code. D was obtained from the PDB, file 1QGT, and colored using Visual Molecular Dynamics (VMD) 1.9.3. This point was added line 136.
8) Small errors
ref 37 and 38 are the same
Ref 38 was removed
line 256 the poor effect of phosphorylation on capsid structure [30, 41, 55, 63]. Ref 63 is not relevant to phosphorylation.
In fact ref 63 was removed.
9) Typos
41 S) essential to promote clatherin mediated endocytosis ‘’clathrin’’ (Line 47).
142 hepatraocellular was removed.
155 (upper raw) was removed.
160 corresponding to c+v+ (green), c+v- (bleu), was removed.
210 S178 was maintained, 5- AAAAAA all photo-acceptors ‘’’ phospho-acceptors ‘’’ was corrected (line 196)
306 IV - 1 - b ARDs endowed with NA chaperon activity ‘’’chaperone’’’ was corrected (line 270)
314 mutants with a decreased, all mutants’’ have’’ decreased in (Line 278)
332 8A, lines 7-11) and found that dissembled HBc ‘’’’ disassembled’’’ (line 296).
477 height among the sixteen R residues remains in HBc ‘’ remain’’ (line 423)
Reviewer 3 Report
This is a very comprehensive well-researched review article.
It requires some editing of the language throughout, as some incorrect or unusual descriptions are used, although it was mostly understandable.
Specific comments: HBeAg is not introduced before Figure 2 legend, please mention in the text.
line 37-39: please clarify polymerase description. RNAseH and TP are not mentioned
line 46: AD38 cells
Figure 1: Can you draw the DNA as partially double stranded, not completely double stranded?
Figure 2: HBeAg should be lowercase g
There are figures throughout the article which are modified original results, such as southern and western blots, taken from other articles (and acknowledged accordingly), eg Figure 5,6,7,8,11. The entire method used and results are then described in the figure legend. Although this is accurate and acknowledged correctly, this level of detail and description of other work is not usually included in a review article, as it requires readers who may not be expert, to extrapolate the meaning. Figure 9 is an example of a Figure more suitable for a review article.
However, this requires clarification with the editors.
Perhaps the authors could consider generating an original figure summarising these findings in illustrative format? This could be included in the last section.
Author Response
Reviewer 3
Comments and Suggestions for Authors
This is a very comprehensive well-researched review article.
Thank you very much for this comment.
It requires some editing of the language throughout, as some incorrect or unusual descriptions are used, although it was mostly understandable.
Specific comments: HBeAg is not introduced before Figure 2 legend, please mention in the text.
HBeAg was introduced line 71.
line 37-39: please clarify polymerase description. RNAseH and TP are not mentioned
These lines were modified to mention these two domains (lines 40-43).
line 46: AD38 cells
This was corrected (line 52)
Figure 1: Can you draw the DNA as partially double stranded, not completely double stranded?
Partially double stranded DNA is now drawn in fig 1 and fig 9.
Figure 2: HBeAg should be lowercase g
This typo was corrected
There are figures throughout the article which are modified original results, such as southern and western blots, taken from other articles (and acknowledged accordingly), eg Figure 5,6,7,8,11. The entire method used and results are then described in the figure legend. Although this is accurate and acknowledged correctly, this level of detail and description of other work is not usually included in a review article, as it requires readers who may not be expert, to extrapolate the meaning. Figure 9 is an example of a Figure more suitable for a review article.
This point was highlighted by the second reviewer. Some figures were removed (5, 7, 11) or partially deleted (6 now n°5 and 10, now n°8). To illustrate our text, figures 8 (now n° 6) and 9 (now n°7) were conserved.
However, this requires clarification with the editors.
This point was emailed and clarified with editors
Perhaps the authors could consider generating an original figure summarising these findings in illustrative format? This could be included in the last section.
Indeed, the § was modified and, in addition, a figure (Fig 9) was added. This figure effectively provides a much better summary of the phosphorylation status and on the location of the CTD during the cycle.
Round 2
Reviewer 2 Report
The authors seem to have considered my requests adequately. The new figure 9 is very welcome as a global summary. It is a bit unclear as is though and should be changed as follows:
'inside the capsid in green and outside the capsid in red. The 549 HBc-CTD phosphorylation and dephosphosphorylation are labelled by + and – marks, respectively '
It's the other way around in the figure as is (red inside, green outside) and the alternate colors are not really necessary (inside/outside are clearly seen without them). But more importantly the +/- distinction is poorly visible. I suggest using the color code instead for this distinction: red phosphorylated, green unphosphorylated. One would thus catch at a glance the correlation hyperphoshorylation/empty particles, that is somewhat obscured by the current presentation.
Author Response
The second reviewer highlighted some details regarding the colors and symbols that illustrate the location and phosphorylation of the CTD. As recommended, Figure 9 was edited with CTDs colored red for phosphorylated CTDs and green for non-phosphorylated CTDs.